# Variation in Gemological Characteristics in Tsavorites with Different Tones from East Africa

Yuanmeng Ma and Ying Guo *

School of Gemmology, China University of Geosciences (Beijing), Beijing 100083, China
* Correspondence: guoying@cugb.edu.cn

**Abstract:** In this paper, the influencing factors of the color and the gemological changes of tsavorites with different tones from East Africa were studied. The gemological characteristics of 35 different green tones in tsavorites were collected based on the results of color measurement, X-ray fluorescence, ultraviolet–visible, infrared and Raman spectroscopy. V and Cr are responsible for the samples' color: with the increase of vanadium content, lightness L* and chroma C* decreased while hue h° increased, and the hue tends to blueish green. The color of tsavorite is related significantly to the absorption bands at about 430 nm and 605 nm through the UV-VIS spectrum. Under long-wave ultraviolet light, the samples show inert or red fluorescence. The G, H, and I peaks of the infrared spectrum are shifted towards the long-wave direction with the reduction of the V content. The peaks at 275 nm, 412 nm and 545 nm on the Raman spectrum tend to move towards the direction of decreasing wavelength with the increase of V content.

**Keywords:** tsavorite; vanadium grossular; gemology; infrared spectrum; Raman spectrum; XRF; color grading

## 1. Introduction

Gem-quality green tsavorite colored by $Cr^{3+}$ and $V^{3+}$ ions is traditionally mined in Tanzania, Kenya and Madagascar in the Neoproterozoic Mozambique metamorphic belt (NMMB). Tsavorite is the gemological trade name for the vanadian grossular discovered by the Scottish geologist Campbell Bridges in north-eastern Tanzania in 1967, about 10–15 km east of the village of Komolo [1–3]. Switzer [4] showed that the green color of grossular is due to the substitution of aluminium by vanadium (and some chromium) in the octahedral site following the formula $Ca_3(Al, V, Cr)_2(SiO_4)_3$, indicating that tsavorite is a solid solution between dominant grossular and minor goldmanite.

Tsavorite is a high-value colored gemstone used in jewelry. This gemstone is durable with a hardness between 7 and 7.5 on the Mohs scale, has no cleavage and is chemically resistant. Its beautiful green color is considered to be more vivid than that of the emerald [5,6]. The factors that determine the considerable value of this gem are the high values of the dispersion (0.028) and refraction (1.740) indices, its excellent saturation, brilliance and transparency and its limited distribution, mostly confined to the Neoproterozoic Mozambique metamorphic belt (NMMB).

Tsavorites are located within metasomatized graphitic rocks such as graphitic gneiss and calc-silicates, intercalated with meta-evaporites. The primary source where tsavorites occur is either in the regional metamorphic nodules (Type I) or in the contact metasomatic quartz vein (Type II), formed in the Pan-African Orogeny movement (650~500 Ma), while sedimentary placers (Type III) represent a secondary source [7].

There are five factors controlling the formation of tsavorites [8–19], i.e., regional lithostratigraphy, lithology, chromophorous elements, high-grade metamorphism, and tectonism. The nodules and quartz veins of primary tsavorite deposits are controlled by lithostratigraphy (especially graphitic gneiss), lithology (especially evaporite), and

structure (especially tectonism), respectively [7]. Tsavorite occurs either as nodules (Type I, including two sub-types: Type $N_I$—initially gypsum and anhydrite concretions, and Type $N_{II}$—initially barite concretions), or in quartz veins (Type II, including three sub-types: Type $II_A$—located at the hinges of the sheared isoclinal folds, Type $II_B$—located within "saddle reef" structures, and Type $II_C$—located within metasomatic zones), or in placers (Type III, including three sub-types: Type $III_A$—eluvial deposits, Type $III_B$—colluvial deposits and Type $III_C$—alluvial deposits) [7].

In this paper, the color of tsavorite was, for the first time, quantitatively characterized on the basis of the uniform color space CIE 1976 L * a * b *; the causes of color and factors that determine it were also investigated. In particular, we investigated the relationships between the colorimetric parameters and the chromophore elements and also the variations in the gemological characteristics of the tsavorite samples, showing different shades through various techniques such as specific gravity, X-ray fluorescence, UV-VIS, IR and Raman spectroscopy.

## 2. Materials and Methods

### 2.1. Samples

In this work, 35 tsavorite gems from the Neoproterozoic Mozambique metamorphic belt (NMMB) in Tanzania and Kenya were analyzed. The gems have faces of similar size, a weight between 0.83 and 1.52 carats (the average weight is 1.15 carats), and a color that varies from pale green to vivid. Since the gems exhibit similar characteristics, we have grouped them as originating from areas of East Africa. Some of the samples are shown in Figure 1.

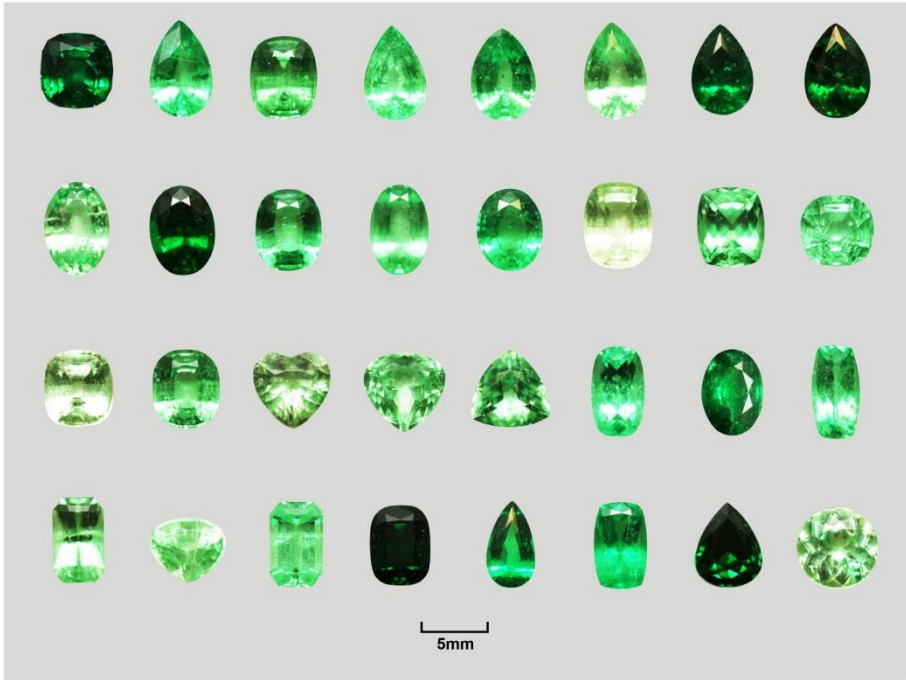

**Figure 1.** Part of the tsavorite sample with different tones used in this study.

Samples were provided by Shaffre Miners Company. Shop address: A53 Shaffre Miners, 1F, Panyu Jewelry City, Panyu District, Guangzhou, Guangdong Province, China. Contact: 18922465632.

### 2.2. X-ray Fluorescence

Energy-dispersive X-ray fluorescence (EDXRF) spectrometry is a quick and non-destructive means of detecting the presence of most elements in samples. Micro-area chemical components were measured by an EDX-7000 energy dispersive X-ray fluorescence

spectrometer (SHIMADZU, Kyoto, Japan) at the jewelry college of China University of Geosciences (Beijing) with the following measurement conditions: atmosphere: air; voltage of 50 kV; 108 μA; 30% DT; collimator (test range), 3 mm, no sample preparation.

### 2.3. Ultraviolet–Visible Spectroscopy

Absorption spectra and transmission spectra in the ultraviolet to visible (UV-VIS) range were recorded with a UV-3600 UV–VIS spectrophotometer (SHIMADZU, Kyoto, Japan) at the jewelry college of China University of Geosciences (Beijing). The test conditions are the following: range of wavelength between 200 and 1000 nm, high scanning speed, sampling interval of 1.0 s, single scanning mode, reflection.

### 2.4. Infrared Spectroscopy and Raman Spectroscopy

The infrared and Raman spectra were collected at the jewelry college of China University of Geosciences (Beijing). We used Bruker tensor 27 Fourier transform infrared spectrometer (BRUKER, Billerica, Massachusetts, USA) as the instrument model. The experimental conditions are as follows: the resolution is 4 cm$^{-1}$, the scanning time of each sample is 50–100, and spectra were collected in reflection geometry.

The Raman spectra were measured with a Horiba LabRAM HR Evolution Raman spectrometer (Horiba Jobin Yvon, Paris, France) equipped with a Peltier-cooled charged-coupled device (CCD) detector, edge filters, and a Nd-YAG laser. The test conditions were as follows: the range of wavelength was 200–2000 cm$^{-1}$; acq.time, 3.0 s; laser, 532 nm—edge.

### 2.5. Colorimetric Analysis

Using different neutral grey backgrounds, an X-Rite SP62 spectrophotometer (X-Rite, Grand Rapids, MI, USA) was used to collect reflective signals from the tsavorites' surface via the integrating sphere. The test conditions can be summarized as follows: CIE standard illumination, D65 light source; reflection, not including the specular reflection; observer view of 2°; measurement range of 400–700 nm. The final color data were averaged three times for testing.

### 2.6. CIE1976 L*a*b* Color System

The CIE1976 L*a*b* color system is composed of the colorimetric coordinates a* and b* and a lightness L*. Chroma C* and the hue angle h° can be calculated using a* and b* as:

$$C^* = \sqrt{a^{*2} + b^{*2}}. \tag{1}$$

$$h° = \arctan\frac{b^*}{a^*}. \tag{2}$$

## 3. Results and Discussion

### 3.1. Gemological and Mineralogical Characteristics

The gemological properties of the 35 studied stones are summarized in Table 1. The saturation of the green color in the samples varies from light to strong, and their diaphaneity ranges from transparent to opaque. RI (refractive index) values fall in the range of 1.729–1.748, and SG (specific gravity) varies between 3.55 and 3.68 g/cm$^3$. Samples are light orange to red under the Chelsea color filter; and the fluorescence under long-wave ultraviolet varies from zero to red, and under short-wave ultraviolet varies from zero to light orange.

**Table 1.** Summary of gemological properties of tsavorites from Kenya and Tanzania.

| Properties | Observation Data |
|---|---|
| Color | Light green to dark green |
| Diaphaneity | Transparent to opaque |
| RI | 1.729–1.748 |
| SG | 3.55–3.68 |
| Fluorescence reaction | Long-wave UV: Inert to red fluorescence<br>Short-wave UV: Inert to light orange fluorescence |

We observe that the refractive index (RI) shows a moderate increasing trend with $w(V_2O_3)/\%$ values (Figure 2 on the left); the relationship is not very linear because the samples belong to the same class of garnets and the refractive indices are therefore very close (between 1.73 and 1.75). An anomalous point with a higher refractive index (1.746) but with a very low V content is noted, while the neighboring samples all have a high V content; this anomaly can be explained by the fact that the sample has a high Cr content, which also influences the index. The refractive index of gemstones is determined by the polarizability of ions in the crystal. The cationic polarizability depends on the ion radius and the structure of the outer electron layer. The greater the radius of cations with the same atomic valence, the higher the polarizability [20]. When vanadium ions replace aluminum ions, the polarizability increases, so the refractive index increases. The refractive index shows a slight positive correlation with the chromium content; the correlation is less evident than that with $V^{3+}$ because the $Cr^{3+}$ content is lower, and its polarizability is also lower than that of $V^{3+}$ [20].

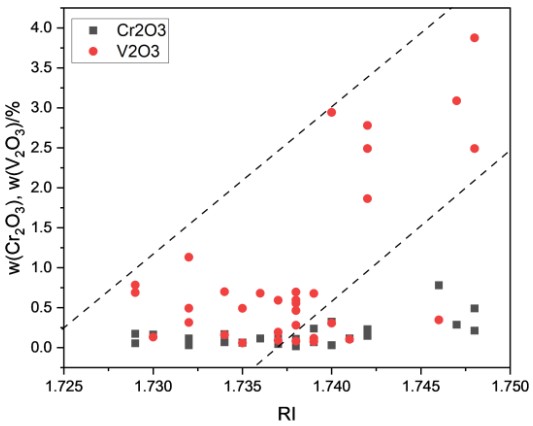 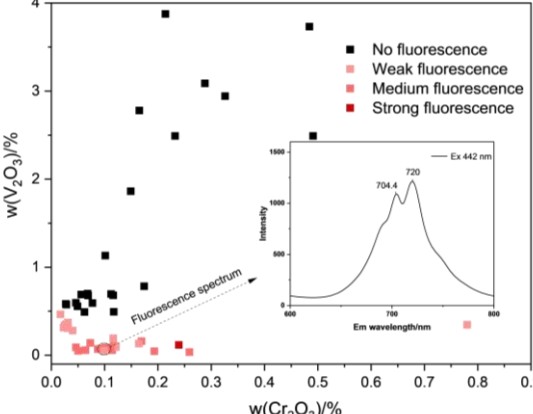

**Figure 2.** Left: Variation between refractive index and $w(Cr_2O_3)$ and $w(V_2O_3)$: RI showing an increasing trend when the $w(V_2O_3)$ increases. Right: Relationship between fluorescence with $w(Cr_2O_3)$ and $w(V_2O_3)$.

The fluorescence (Figure 2, right) is related to both chromium and vanadium. We can clearly observe that when the vanadium content is greater than a certain amount (about 0.5% in this article), the tsavorite does not fluoresce, while when the vanadium content is lower, the gem shows a red fluorescence. Furthermore, it is observed that with the increase of the chromium content in the gems, the red fluorescence also increases. Figure 2 on the right shows the inhibition of vanadium on the effect of fluorescence; strong, medium and weak fluorescence is evaluated with the naked eye. The graph analyses indicate that chromium is the main source of fluorescence, while vanadium suppresses it. We can also learn from the fluorescence spectrum of a sample with red fluorescence that when the excitation light source is 442 nm blue-violet light, the strongest emission light source is 720 nm red light.

### 3.2. Color Quantification and Classification

Using an ultraviolet-visible spectrophotometer, the colors of the 35 tsavorites samples were measured to obtain their values of lightness L* (27.8–64.7), colorimetric coordinates a* (−27.8 to −2.8) and b* (−6.4–13.7), chroma C* (6.99–30.99), and hue angle h° (152–246.4). These values are consistent with the color appearance of tsavorites. The color parameters of the 35 samples are projected in the CIE 1976 L*a*b* uniform color space (Figure 3a).

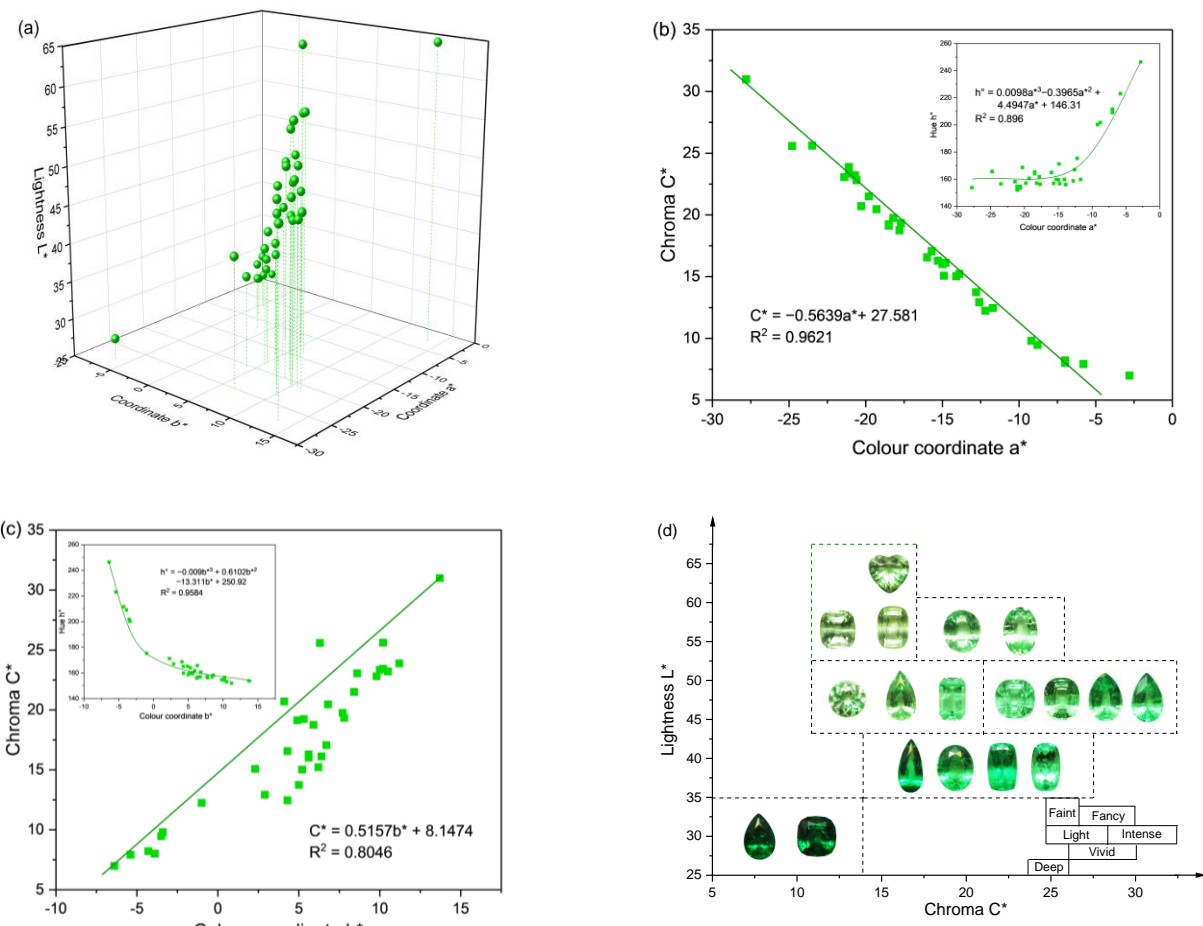

**Figure 3.** (**a**) 35 tsavorites plotted in CIE 1976 L*a*b*uniform color space; (**b**) correlation between color coordinate a* with chroma C* and hue h°; (**c**) correlation between color coordinate b* with chroma C* and h°; (**d**) tsavorites' color grading is divided into: deep, vivid, intense, fancy, light, faint.

Through the analysis of the color parameter of 35 tsavorites, it is observed that the correlation between the color coordinate a* with chroma C* (Figure 3b, $R^2$ = 0.962) is greater than that between b* with C* (Figure 3c, $R^2$ = 0.804), so the chroma of tsavorite is mainly controlled by a*. Furthermore, we observe that the correlation between the color coordinate b* with hue angle h° (Figure 3c, $R^2$ = 0.958) is greater than that between a* with hue angle h° (Figure 3b, $R^2$ = 0.896), so the hue of tsavorite is mainly controlled by b*.

Based on the CIE 1976 L*a*b* uniform color-space system, K-means cluster analysis and Fisher discriminant analysis were used to classify the tsavorites' colors. These two methods have been verified for classifying the colors of gemstones, such as peridot [21], tourmaline [22], jadeite-jade [23–27] and amethyst [28], among others. K-means cluster analysis is a statistical analysis technique that divides the research object into relatively homogeneous groups. As one of the most important methods in multivariate statistical analysis, Fisher discriminant analysis summarizes the regularity of all types of samples and establishes discriminant formulae and criteria to identify the types of new sample points according to the information provided by known sample categories [21].

The color parameters L*, a* and b* were classified by K-means cluster analysis. In this paper, we tested the accuracy of three to seven classes of clustering results, respectively, (Table 2) and found 100% accuracy when they are divided into six classes, so six classes were used as the final classification standard.

**Table 2.** The clustering accuracy corresponding to different clustering numbers.

| Clustering Number | 3 | 4 | 5 | 6 | 7 |
|---|---|---|---|---|---|
| Error number | 1 | 2 | 1 | 0 | 2 |
| Accuracy | 97.14 | 94.29 | 97.14 | 100 | 94.29 |

The color parameters L*, a* and b* were entered into the Fisher discriminant formulae. From the results of the K-means cluster analysis and Fisher discriminant analysis and by imitating the colored-diamond grading system of the Gemological Institute of America [29], the tsavorites' color is divided into the grades of (1) Faint, (2) Light, (3) Fancy, (4) Fancy Intense, (5) Fancy Vivid, and (6) Fancy Deep. The tsavorites' color grading system so established is shown in Figure 3d.

### 3.3. Chemical Composition Analysis

Energy-dispersive X-ray fluorescence (EDXRF) spectrometry is a quick and non-destructive means of detecting the presence of most elements in tsavorites. The results show that the range of weight percentage [wt%] of the main oxide in the 35 tsavorites samples is $w(SiO_2)$ = 34.24–41.97, $w(CaO)$ = 33.1–38.09, $w(Al_2O_3)$ = 18.43–23.82, while those of the other oxides are $w(V_2O_3)$ = 0.059–3.876, $w(Cr_2O_3)$ = 0.017–0.78, $w(MnO)$ = 0.22–1.32, $w(Fe_2O_3)$ = 0.052–0.116, $w(CuO)$ = 0.001–0.021. In the detected elements, V, Cr and Fe are transition-metal elements, which are often related to the color of gemstones. Bivariate correlation analysis was used to analyze the correlation between the contents of transition-metal elements and the color parameters of the 35 tsavorite samples (Figure 4).

The results show a high negative correlation between $w(V_2O_3)$ and chroma C* and lightness L* in tsavorites ($R^2_{C*}$ = 0.826, $R^2_{L*}$ = 0.8667, when $R^2$ is closer to 1, there is a higher correlation) (Figure 4a,b), and high positive correlations with hue angle h° ($R^2_{h°}$ = 0.886) (Figure 4c). There is no significant correlation between $w(Cr_2O_3)$ and $w(Fe_2O_3)$ with chroma C*, lightness L* and hue angle h°. With the increase of $w(Cr_2O_3)$, the hue angle h° shows an increasing trend, the lightness L* a decreasing trend, and chroma C* first falls and then rises.

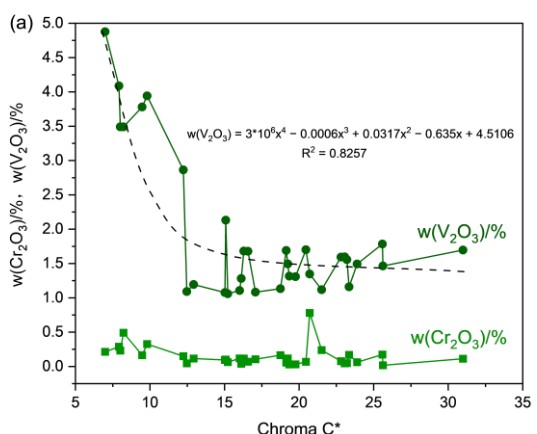
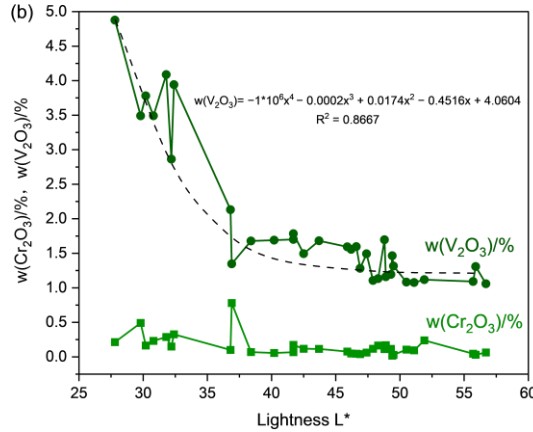

**Figure 4.** *Cont.*

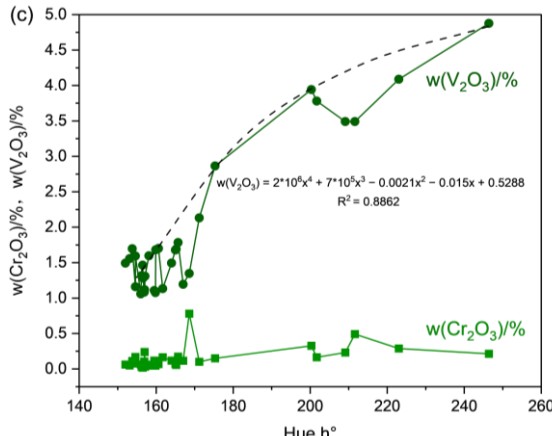

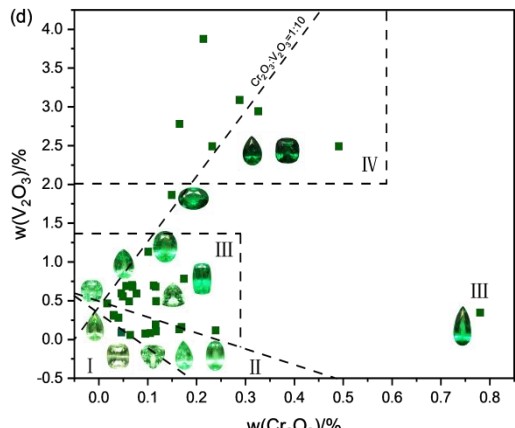

**Figure 4.** (**a**) Correlation between chroma C* and w($Cr_2O_3$) and w($V_2O_3$); (**b**) correlation between lightness L* and w($Cr_2O_3$) and w($V_2O_3$); (**c**) Correlation between hue h° and w($Cr_2O_3$) and w($V_2O_3$); (**d**) the content range of Cr and V elements of tsavorite in different colors, which is divided into four areas according to the visual effect: light green (I), mint green (II), intense green (III) and dark green (IV).

Mapping the element content of 35 samples with the chromatic data revealed a high correlation between w($V_2O_3$) and color parameter L*, C*, h°. We also tried to interpret the concentrations of Cr and V at particular points on the graphs. The convex point of the w($Cr_2O_3$) line corresponds to the concave point of the w($V_2O_3$) line, which means that the chromium element plays a supplementary role to the vanadium element. So, in tsavorites with a higher V than Cr content, the effects of these elements on hue are consistent, but V plays a dominant role over Cr. The hue angle increases with increasing V content, and consequently the hue tends towards bluish-green.

We believe that by checking the Cr and V contents we can obtain the samples with the best gemological properties, which are those in region III (Figure 4d). We cannot partition the rich Cr zone because we lack the Cr-rich samples. When making synthetic gemstones, if you want a vivid green color, you can refer to the Cr and V contents of area III, while if you want a fresh mint green color, you can refer to the Cr and V contents in area II. Instead, we should avoid the compositions of areas I and IV, as the samples here are too light or too dark.

The samples under long-wave ultraviolet radiation produced red fluorescence because of $Cr^{3+}$, which gradually weakens and disappears as the V content increases, indicating that V is an important fluorescence quencher (see Figure 5). When the excitation light source is 420 nm, the emission spectra appear predominantly as fluorescence peaks at 704.4 and 720 nm, which is caused by the multiple forbidden resistance transition $^2E_g{\rightarrow}^4A_{2g}$ of $Cr^{3+}$. The existence of two effective excitation bands is related to the two spins of $Cr^{3+}$ allowing absorption transitions, with the purple light (400–440 nm) region absorption resulting from $^4A_{2g}{\rightarrow}^4T_{1g,}$, and the yellow region (550–600 nm) from $^4A_{2g}{\rightarrow}^4T_{2g}$ [30]. The $Cr^{3+}$ alternative of $Al^{3+}$ in garnet occupies the octahedral center and is in the octahedral strong field, and with the excited state level $^2E_g$ below $^4T_{2g}$, the electron absorption energy moves from ground state $^4A_{2g}$ to excited state $^4T_{1g}$, to the lowest excited state $^2E_g$ level by vibration relaxation or other radiation-free transitions, and then from $^2E_g$ to the ground state $^2E_g$. The fluorescence yield at the time is strongest when the energy of the source can allow these two absorption transitions to occur [31].

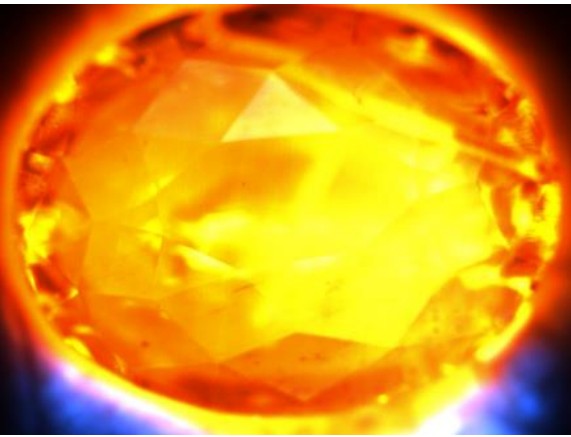

**Figure 5.** Some samples show orange-red fluorescence under long-wave ultraviolet.

### 3.4. UV-VIS Spectrum Analysis

The UV-VIS spectra of tsavorites are shown in Figure 6. The UV-VIS spectrum of the tsavorites sample shows that there are two obvious absorption bands at about 430 nm and 605 nm. As 520 nm is the transmission window, the tsavorites are green. With the increase of the transmission window, the hue angle decreases and the hue changes to yellow-green.

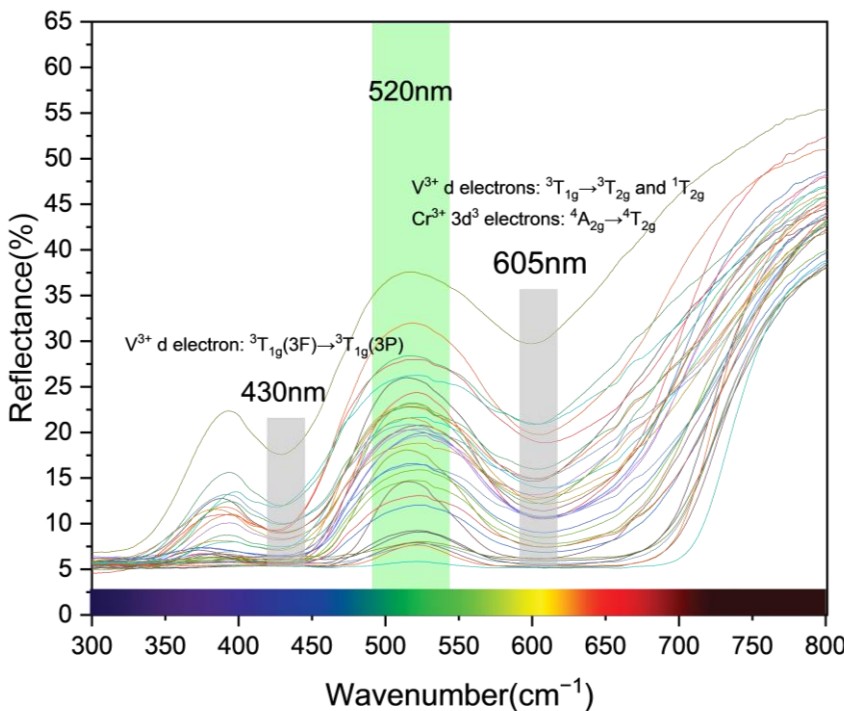

**Figure 6.** UV-VIS spectrum of 35 tsavorites (520 nm is the transmission window).

In tsavorites with high V content, the absorption at 430 nm is due to V, while in those rich in Cr rather than V, there are two additional weak peaks at 697 and 701 nm [32]. As the Cr content increases, these peaks become stronger, but they disappear when the V content is higher than Cr. The XRF spectra show that the V content of the tsavorites sample is higher than that of Cr. The absorption band at 430 nm is related to the substitution of $Al^{3+}$ by $V^{3+}$ in the coordination octahedron. The d electron of $V^{3+}$ transitions between $^3T_{1g}$ (3F) and $^3T_{1g}$ (3P) levels to produce this absorption. The 605 nm consists of 590 nm and 643 nm. The absorption peak at 643 nm is attributed to the transition of d electrons of $V^{3+}$ from $^3T_{1g}$ ground state to the $^3T_{2g}$ and $^1T_{2g}$ states. The absorption peak at 590 nm can be attributed to the transition absorption of the $3d^3$ electrons of $Cr^{3+}$ between $^4A_{2g}$ and $^4T_{2g}$

energy levels. The absorption peak at about 385 nm is related to Fe. Garnet can also be colored by charge transfer, such as $Fe^{2+}$–$Ti^{4+}$ and $Fe^{2+}$–$Fe^{3+}$. For tsavorite, only $Fe^{2+}$–$Ti^{4+}$ charge transfer needs to be considered, which is confirmed by the fact that the content of Ti is higher than that of Fe.

We can see that the area of the 520 peaks decreases as the color darkens (Figure 7a). We observe that the height of the 520 nm peak (Figure 7b) and its area (Figure 7c) show extremely similar relationships with hue h°, chroma C*, and lightness L*. They have a high negative correlation with h°. As the height and area of the 520 nm peak increases, the hue of the gem decreases and becomes yellowish green. There is a medium-high positive correlation between the height and the area of 520 nm peak with L* and C*. The lightness and chroma increase with the increase of the height and area of 520 nm peak.

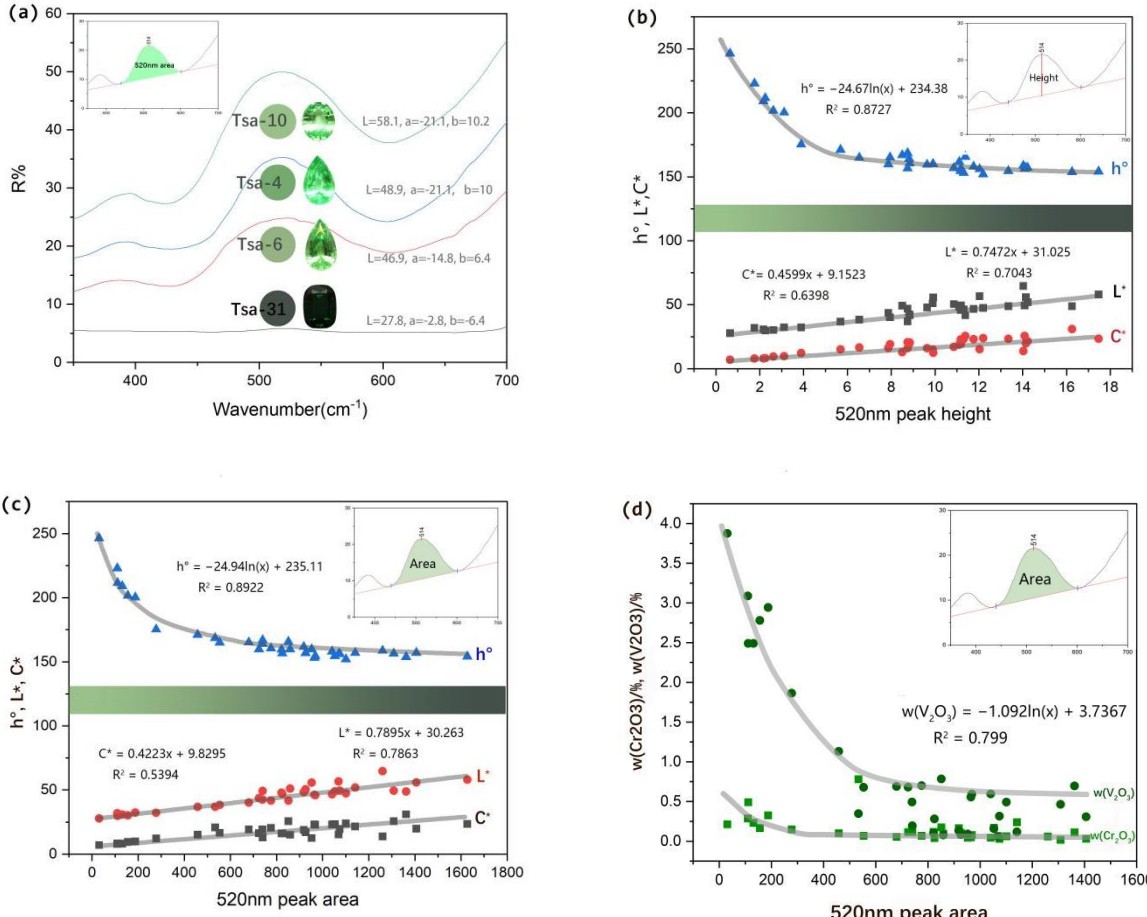

**Figure 7.** (**a**) The 520 nm peak changes with samples with different hues (R% = Reflectance%); (**b**) the changes of L*, C*, h° with the increase of 520 nm peak height; (**c**) the changes of L*, C*, h° with the increase of 520 nm peak area; (**d**) the changes of w($V_2O_3$) and w($Cr_2O_3$) with the increase of 520 nm peak area.

Figure 7d shows that with the increase of the 520 nm peak area, the V element shows an obvious downward trend, and the Cr element shows a weak downward trend. It indirectly shows that with the decrease of V and Cr content, L* and C* increase and h° decreases.

### 3.5. Infrared Spectroscopy

The infrared spectrum (Figure 8) of the tsavorite sample shows that the A and B peaks form a wide absorption band of 940–965 cm$^{-1}$. The peaks A, B, C, D, E, F, G and H are at $964 \pm 5$ cm$^{-1}$, $943 \pm 3$ cm$^{-1}$, $869 \pm 3$ cm$^{-1}$, $844 \pm 3$ cm$^{-1}$, $619 \pm 1$ cm$^{-1}$, $557 \pm 2$ cm$^{-1}$, $511 \pm 1$ cm$^{-1}$, $489 \pm 3$ cm$^{-1}$ and $459 \pm 3$ cm$^{-1}$.

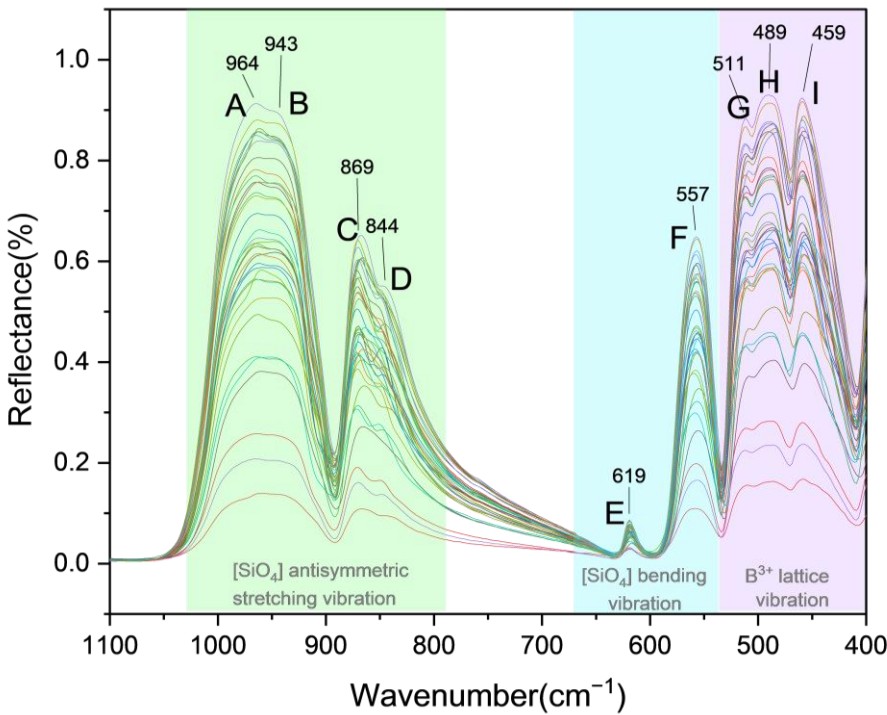

**Figure 8.** Infrared spectrum of 35 tsavorites.

The A, B, C, D, E, and F absorption peaks are related to the relevant vibration inside the $[SiO_4]^{4-}$ group. The G, H, and I absorption peaks are generated by lattice vibration, that is, the vibration of other cations except Si ions. A. M Hofmeister et al. [33] believe that the absorption peak in the range of 400–500 $cm^{-1}$ is formed by the motion of trivalent cations entering the $[BO_6]$ lattice (Table 3). In this paper, the trivalent cations are mainly $Cr^{3+}$ and $V^{3+}$. In order to study the effect of chromogenic elements on infrared spectra, we focus on the change of G, H, and I peaks. Since V plays a major role in defining color (when present in high levels), we first studied the influence of this element on the infrared spectrum. By selecting seven samples with increasing content of V, we observe that with decreasing content of this element, the G, H, and I peaks shift in the direction of increasing wavelength.

**Table 3.** Chromium–vanadium grossular infrared spectrum peak assignment.

| $[SiO_4]^{4-}$ Internal Vibration | | | | | | Lattice Vibration | | |
|---|---|---|---|---|---|---|---|---|
| ν3(antisymmetric stretching vibration) | | | | ν4, ν2(bending vibration) | | Related to $B^{3+}$(trivalent cation) vibration | | |
| A | B | C | D | E | F | G | H | I |
| $964 \pm 5\ cm^{-1}$ | $943 \pm 3\ cm^{-1}$ | $869 \pm 3\ cm^{-1}$ | $844 \pm 3\ cm^{-1}$ | $619 \pm 1\ cm^{-1}$ | $557 \pm 2\ cm^{-1}$ | $511 \pm 1\ cm^{-1}$ | $489 \pm 3\ cm^{-1}$ | $459 \pm 3\ cm^{-1}$ |

Then, by selecting samples with the same amount of V, we can observe the influence of Cr on the infrared spectrum, but the influence of Cr does not follow the same unified law with V (Figure 9b–d). Most cases show that with the increase of Cr, some peaks in G, H, and I move in the direction of increasing wavelength, but one case moves significantly differently in the direction of decreasing wave number (Tsa-20 and Tsa-27). It may be that the measurement of XRF data is not accurate enough to accurately reflect the slight change of content.

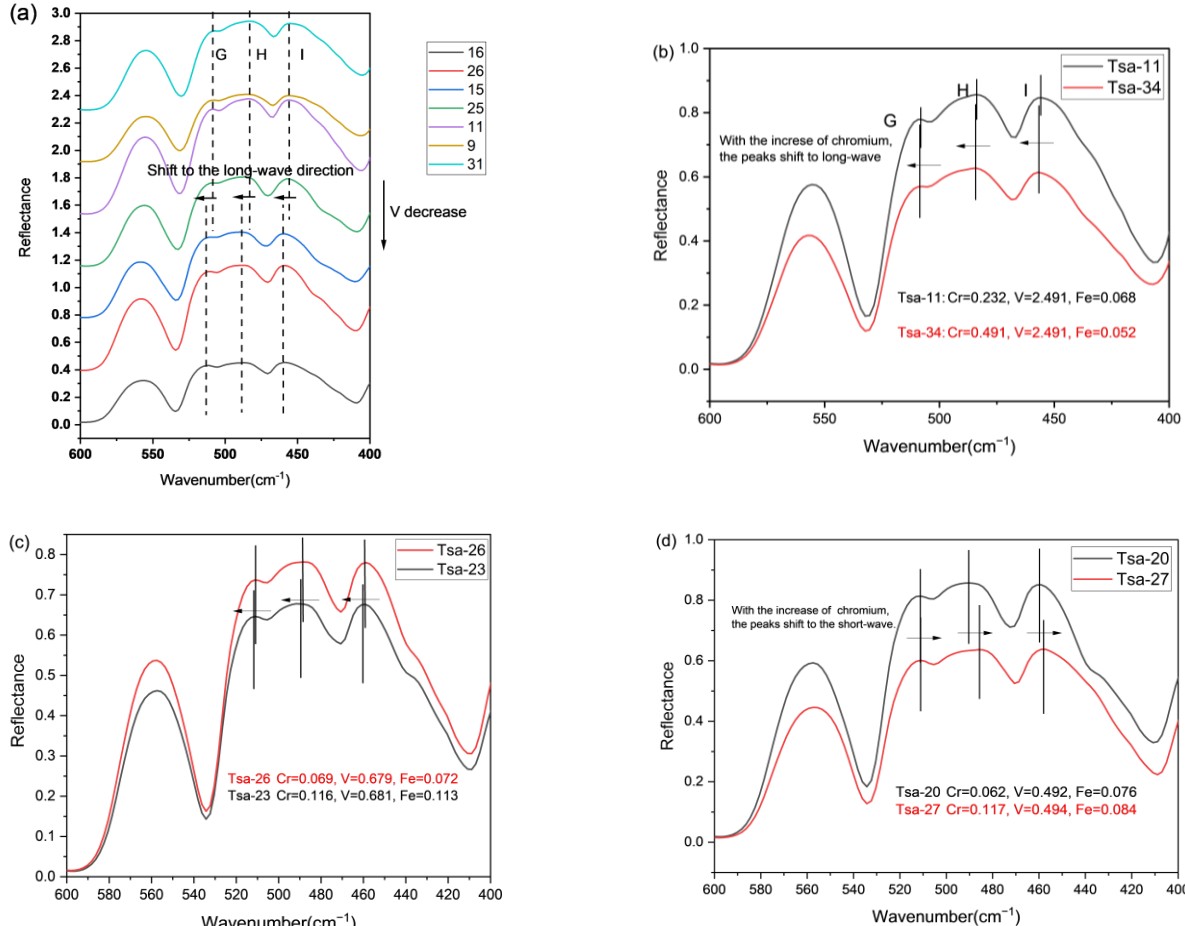

**Figure 9.** (**a**) With the decrease of vanadium, the peaks shift to long wave; (**b**,**c**) with the increase of chromium, the peaks shift to long wave; (**d**) with the decrease of chromium, the peaks shift to long wave.

### 3.6. Raman Spectroscopy

The Raman shift of the $A_{1g}$ mode belonging to $(Si-O)_{str}$ appears near 878 cm$^{-1}$, and that of $F_{2g}$ mode is near 821 cm$^{-1}$. The Raman shift range of the $A_{1g}$ mode belonging to $(Si-O)_{bend}$ is 545 cm$^{-1}$, and the Raman shift of $E_g$ mode appears near 493 cm$^{-1}$. The peak intensity of the $A_{1g}$ mode belonging to $R(SiO_4)^{4-}$ is high, and the Raman shift range is about 371 cm$^{-1}$. The Raman shifts of the $T_{2g}$ mode belonging to $T(X^{2+})$ range from 275 cm$^{-1}$ to 245 cm$^{-1}$, and the Raman shifts of the $E_g$ mode belonging to $T(SiO_4)^{4-}$ appear near 180 cm$^{-1}$ (Table 4) [32].

**Table 4.** Chromium–vanadium grossular Raman spectrum peak assignment.

| Si-O$_{Str}$ | | Si-O$_{Bend}$ | | R(SiO$_4$)$^{4-}$ | | T(X$^{2+}$) | T(SiO$_4$)$^{4-}$ |
|---|---|---|---|---|---|---|---|
| A$_{1g}$ | E$_g$ + F$_{2g}$ | A$_{1g}$ | E$_g$ + F$_{2g}$ | A$_{1g}$ | E$_g$ + F$_{2g}$ | E$_g$ + F$_{2g}$ | E$_g$ + F$_{2g}$ |
| 878 | 1005, 821 | 545 | 493, 507, 626 | 371 | 412 | 245 275 | 180 |

The $A_{1g}$ Raman active mode of tsavorite contains 371, 545 and 878 cm$^{-1}$ absorption peaks, which can be assigned to $R(SiO_4)^{4-}$ ($[SiO_4]^{4-}$ tetrahedral rotational vibration), Si-O$_{Bend}$ (Si-O bridge oxygen bending vibration) and Si-O$_{str}$ (Si-O non bridge oxygen stretching vibration), respectively. The frequency of $T(SiO_4)^{4-}$ ($[SiO_4]^{4-}$ tetrahedral translational vibration) should be $1.5 \pm 2$ times lower than that of $R(SiO_4)^{4-}$, because the vibration of $T(SiO_4)^{4-}$ consists of the whole tetrahedral unit, while $R(SiO_4)^{4-}$ only involves oxygen

anions. Only one peak of the $E_g$ Raman active mode was observed in the frequency range less than 200 cm$^{-1}$. Therefore, this peak was assigned to $T(SiO_4)^{4-}$. In the Raman spectra of green garnet samples, two peaks of $F_{2g}$ Raman active modes are observed at the low frequency of 200–300 cm$^{-1}$, which are assigned as $T(X^{2+})$ (translational vibration of divalent cations). The splitting of these two modes is the result of two different force constants acting on the oxygen atoms in the X–Y bonding plane of the Ca and X positions.

The peaks at 275 cm$^{-1}$, 412 cm$^{-1}$ and 545 cm$^{-1}$ tend to move towards the direction of decreasing wavelength with the increase of the V element (Figure 10). That is, the Raman shift attributed to Si-O$_{Bend}$ shifts to the short-wave direction with the increase of the B$^{3+}$ ion radius, indicating that with the increase of the B$^{3+}$ ion, [BO$_6$] the octahedron also increases accordingly, which reduces the bond energy of the Si-O chain connected with it, resulting in the short-wave direction shift of Raman shift attributed to Si-O$_{Bend}$. The other two shifts are also due to the decrease of bond energy.

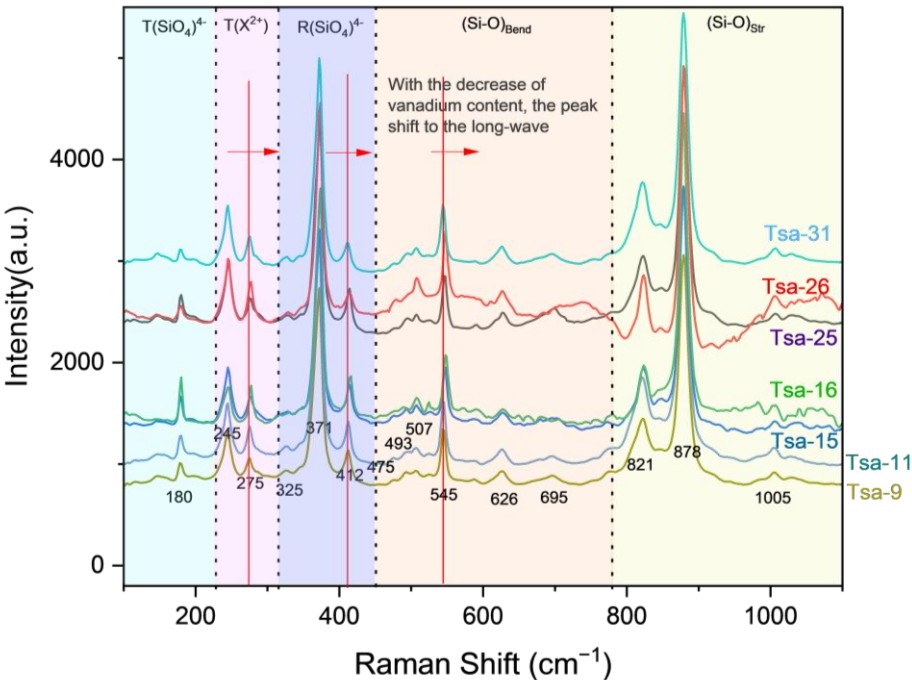

**Figure 10.** Raman spectroscopy of 7 tsavorites. The order of vanadium content from small to large is: Tsa-16, Tsa-26, Tsa-15, Tsa-25, Tsa-11, Tsa-9, Tsa-31.

## 4. Conclusions

This work showed that chromium and vanadium are the main elements that determine the green color in tsavorites. In samples where the vanadium content is greater than that of chromium, V plays a dominant role in determining the color, while Cr is auxiliary. We observed a high negative correlation between the lightness L* and the chroma C* with V content, and high positive correlation between the hue angle h° with the V contents. As the V content increases, the hue angle also increases, and as a result the hue tends to a bluish-green color. The UV-VIS spectra of the tsavorite sample show that there are two obvious absorption bands at about 430 nm and 605 nm. The wavelength of 520 nm is the transmission window; when it increases, the hue angle is decreased and the hue shifts to yellowish-green. Infrared spectroscopy showed that with decreasing V content, the G, H, and I peaks shift in the direction of increasing wavelength. The peaks at 275 nm, 412 nm and 545 nm on the Raman spectrum tend to move towards the direction of decreasing wavelength with the increase of V.

**Author Contributions:** Conceptualization, Y.M.; methodology, Y.M.; software, Y.M.; validation, Y.M. and Y.G.; formal analysis, Y.M.; investigation, Y.M.; resources, Y.M.; data curation, Y.M.; writing—original draft preparation, Y.M.; writing—review and editing, Y.M.; visualization, Y.M.; supervision, Y.G.; project administration, Y.M.; funding acquisition, Y.G. All authors have read and agreed to the published version of the manuscript.

**Funding:** The APC was funded by the Institute of Jewelry, China University of Geosciences, Beijing and Professor Ying Guo (guoying@cugb.edu.cn).

**Institutional Review Board Statement:** Not applicable.

**Informed Consent Statement:** Not applicable.

**Data Availability Statement:** Not applicable.

**Acknowledgments:** The experiments in this article were conducted in the laboratory of the Gemological Institute, China University of Geoscience, Beijing. I would like to thank Chen Jinqian of Tsavorite Miner for providing all the samples.

**Conflicts of Interest:** The authors declare no conflict of interest.

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
