# Peer review of "Variation in Gemological Characteristics in Tsavorites with Different Tones from East Africa"

_crystals, doi:10.3390/cryst12111677_

Round 1

Reviewer 1 Report

RE: Variation of gemmological characteristics of tsavorite (from East Africa) in different tones

This manuscript provides important and comprehensive information on the features and gemological characteristics of tsavorite in different tones from East Africa. This manuscript is very well organized and has a high quality of writing. I recommend this manuscript with minor revision for acceptance and publication. Some necessary edits before accepting the manuscript for publication are:

1- In the whole manuscript, for oxides such as Cr2O3, numbers should be shown as subscripts.

2- Captions of tables and figures are stated very briefly. It is better to write the captions of tables and figures in more detail.

Author Response

Thank you for your comments and suggestion concerning our manuscript.

Here are my responses to the questions:

1- In the whole manuscript, for oxides such as Cr2O3, numbers should be shown as subscripts.

Re:  Thanks, I have checked and corrected one by one.

2- Captions of tables and figures are stated very briefly. It is better to write the captions of tables and figures in more detail.

Re:  Thanks, I have completed all the captions of tables and figures.

Reviewer 2 Report

This is a thorough gemmological study of tsavorites with established methods. The presentation is overall clear and sufficiently detailed without unneccessary excursions. I recommend publication after addressing the following minor points:

 green tones in tsavorites were studied based 7
8: spectrum -> spectra

Beautiful green tsavorite -> Gem-quality green tsavorite  (no qualifications like 'ugly' or 'beautiful' in scholarly papers!)

his article chose 54 faced tsavorite gems from Tanzania and Kenya as samples : Please provide information about the provenance: Bought/loaned from whom? Where are they located - which museum or research/private collection, if available: accession number.

65: vX-ray fuorescence spectrometer with the measure conditions as follows -> Chnage to: X-ray fLuorescence spectrometer with the following measureMENT conditions

 collimator, 5 mm. : Please rephrase for clarification.

The infrared spectrum test site is the jewelry College of China University of Geosci- 73
ences (Beijing), and the instrument model
-> The infrared spectra were cllection at the... . We used... as instrument model.

the detection mode is reflec- 76
tion mode. -> Spectra were collected in reflection geometry.

Raman: what spectram resolution?

edge 79
flters -> You mean notch filter?

It shows that chromium is the main source of fluorescence and vanadium is the 115
quencher of fluorescence.
-> ... and Vanadium suppresses the fluorescence.
Can you provide an explanation? (Charge transfer?)

It is known from XRF that the V content of 223
tsavorites sample is higher than that of Cr.
-> The XRF spectra show that the V content ....

hen select samples with the same amount of V to observe the influence of Cr on the 267
infrared spectrum, but the influence of Cr on it does not reach the same unified law.
-> ??? Do you mean: We selected samples with equal amount of V to observe... but the effect of Cr did not confirm above (?) correlation?

he peaks at 275nm, 412nm and 545nm tend to move towards the direction of de- 306
creasing wavelength with the increase of V element (Figure 18).
-> a) you mean Figure 10?
-> b) cm^-1 NOT nm!!!

Author Response

Thank you for your comments and suggestion concerning our manuscript. Here are the responses to the questions:

  1. Re: Thanks, I have changed the word "spectrum" to "spectra".
  2. Re: Thanks, I have changed "Beautiful green tsavorite" to "Gem-quality green tsavorite".
  3. Re: Samples were provided by Shaffre Miners Company. Shop address: A53 Shaffre Miners, 1F, Panyu Jewelry City, Panyu District, Guangzhou, Guangdong Province, China. Contact: 18922465632.
  4. Re: Thanks, I have changed "X-ray fluorescence spectrometer with the measure conditions as follows" to: "X-ray fluorescence spectrometer with the following measurement conditions".
  5. Re: "Collimator" means test range, I add it in the article.
  6. Re: Thanks, I modified the sentence "The infrared spectra were collected at the...".
  7. Re: Thanks, I changed the sentence to "Spectra were collected in reflection geometry."
  8. Re: Spectra resolution is the minimum wavelength interval to detect spectral radiation energy, and to be precise, is the spectral detection capability.
  9. Re: Yes, I mean edge filters, I lost "i". It can reduce spectral noise in Stokes shift applications
  10. Re: The reason of fluorescence quenching may be the weak binding between ground state fluorescence molecules and quenching agent to form complex or the energy transfer or physical collision between excited state fluorescence molecules and quenching agent. There is no exact mechanism of fluorescence quenching in the literature. The specific mechanism of fluorescence inhibition needs further detailed study.
  11. Re: Thanks, I revised the sentence.
  12. Re: Yes, the effect of Cr did not confirm above correlation, because the content of Cr is too low and the accuracy of XRF data can not be reached to two decimal places.
  13. Re: Thanks, I corrected the problems.

Reviewer 3 Report

Please see comments on the paper. details regarding the aim of this work are missing and gemstones description is needed to set the scene. All the figures need to be looked at and improved to give the reader a better understanding, some numbering are wrong and some figures are not recalled in the main text at all! more info are needed and better figure cations are required.

Author Response

Dear reviewer:

Thanks for your detailed comments and suggestions.

I have answered your questions one by one and added them to the text. All the mistakes have been corrected, and the corrections and additions have been colored yellow. The final experimental sample number of this article is 35, and I have revised all of them.

Thank you very much for your help.

Sincerely,

Yuanmeng Ma

[email protected]

Reviewer 4 Report

The work “Variations of  gemological characteristics of tsavorites (from East Africa), in different tones”,  have several important critical issues, and unfortunately on my opinion cannot  be accepted for publication.

In the introduction, the motivations and aims of this research and the strategy used to achieve it must be explained in detail.

Paragraph. 2: How do the authors be sure that the analyzed samples are actually tsavorites? Are there any chemical or diffractometric analyses to demonstrate it? If they were done, the authors could mention them. Alternatively, are optical and gemological analyzes sufficient for this purpose? Also, the authors should explain how they got the samples. Did they buy them in any market?

The sub-paragraph 2.2 on X-Ray  fluorescence is very limited :the techniques of preparation of the sample, whether it was powdered or not, and the analytical conditions need to be explained in more detail. Furthermore, it is necessary to specify in which laboratory these analyzes were carried out and which was the model of instrument used. Also for ultraviolet spectroscopic analyzes, the analytical methodologies, the model of instrument used, the laboratory where the analyzes were conducted must be explained in greater detail.

Figures 2, 3, 4, 7 which show the various correlations are very small and their reading is very difficult. Fig. 2: on my opinion the correlation between the refractive index and Cr does not exist.

In paragraph 3.3 numerous correlations are reported between the descriptive parameters of the color but no explanation of these trends is provided.

Paragraph 3.3: Chemical analysis.  I  think this is the main problem of the work . The authors state that this technique was used only to obtain qualitative and not quantitative data, and in fact no table of values is reported. Consequently, I believe that all correlations and discussion of the data presented in subsequent discussions have no real meaning

Author Response

Dear reviewer:

Thanks for your comments and suggestions, they are really helpful.

Here are my response to your questions:

  1. Re:  I added the aims of this research and the strategy used to achieve it in the abstract and the introduction: "In this paper, the color of tsavorite was characterized quantitatively based on the CIE 1976 L*a*b* uniform colour space for the first time, and we discussed the causes of tsavorite colour and the influence factors. We study the effect of transition metals on colour through the UV-VIS spectra, try to find the relationship between composition with colour. The changes of gemmological characteristics of tsavorite in different tones (including specific gravity, UV fluorescence, chemical omposition, uv-vis spectra, IR spectra and Raman Spectra) was also studied. "
  2. Re: The infrared data and the Raman data have maps corresponding to standard samples, which can directly prove the gemstone type. I attach a report of the analysis of the Raman map in the annex. And samples were provided by Shaffre Miners Company. Shop address: A53 Shaffre Miners, 1F, Panyu Jewelry City, Panyu District, Guangzhou, Guangdong Province, China. 
  3. Re: No sample preparation is required, XRF is a quick and non-destructive means of detecting the presence of most elements. The other detail of  instrument (condition, laboratory, model) I have added to the manuscript.
  4. Re: I have redescribed figure 2 in the article: "We find that refractive index (RI) showing an increasing trend when the w(V2O3)/% goes up (Figure 2 left). They haven’t form an obvious linear relationship because they belong to the same garnet class, grossular, in which the refractive indices of gemstones are very close. The difference in the content of V is not large enough to cause a significant change in the refractive index. The refractive index changes within a very small range (1.73-1.75) , so the graph shows an upward trend over a wide range rather than a linear relationship. We can also find a special point whose refractive index is 1.746. Its V content is very small but it has a high RI, while the nearby samples all have the high V content. That is because the Cr content of the special point is obviously higher than other samples, the result shows that the influence of Cr element on refractive index is greater than that of V element. " The chromium content is so low that there is no obvious correlation in the figure, but it can be proved that the chromium content is related to the refractive index at the special points.                                        I think Figure 3 have a high correlation, whose R^2=0.96 and R^2=0.8 (they are close to 1). Figure 4 also has a high correlation with all R^2>0.8. (Cr content is too low to show the correlation, but it is meaningful to study some special points. ). In Figure 7, lightness and hue have a high correlation with 520nm peak area, chroma has a medium correlation with 520nm peak area. These figures are all scatter plot with a trend line, they are easy to read.
  5. Re: I added an explanation of the correlation between the color parameters.
  6. Re: I wrote wrong, XRF is a semi-quantitative non-destructive test, not only qualitative analysis, I added the content data in manuscript:"w(SiO2)=34.24-41.97, w(CaO)=33.1-38.09, w(Al2O3)=18.43-23.82, and those of the other oxides are w(V2O3)=0.059-3.876, w(Cr2O3)=0.017-0.78, w(MnO)=0.22-1.32, w(Fe2O3)=0.052-0.116, w(CuO)=0.001-0.021."

Round 2

Reviewer 4 Report

I have verified that the authors have satisfied, at least in part, my perplexities for which I did not consider the work suitable for publication. I also acknowledge that three reviewers felt that the first version of the work could be accepted with only minor changes. However, there are still some points on which the authors have to work, to make the work acceptable. The first problem is the English form which is still inadequate, with numerous errors and unclear sentences. In this regard, I worked hard to improve the English form of the work and the corrections are included in the attached PDF file. I invite the authors to consider them, even if I don't know if they are sufficient. Another question concerns the presence of Iron, which is a possible chromophore element. How can the authors be sure that it does not exert an influence on the color of the tsavorites, perhaps because in low quantities? This point should be explained better.

Also:

Lines 245-247: “The UV-Vis spectrum of tsa-245 vorites sample shows that there are two obvious absorption bands at about 430nm and 246 605nm” Why obvious? Are both the absorption bands related to Vanadium? If so then write it down

Figure 7A and line 265: please indicate the meaning of R% and Tsa.

Author Response

Thank you for your careful revision of the English form of the article, they must have spent a lot of your energy, I have revised all the sentences. The revised article read more concise and clear.

Here are my responses to the questions:

  1. Q: Another question concerns the presence of Iron, which is a possible chromophore element. How can the authors be sure that it does not exert an influence on the color of the tsavorites, perhaps because in low quantities? 

         R: I wrote in 203 "There is no significant correlations between w(Cr2O3) and          w(Fe2O3) with chroma C*, lightness L* and hue angle h°."(After the relevant          data mapping.) Yes, this because the iron content is too low.

     2.  Re: In 256-269, the formation reasons of two absorption bands (430 and              605nm) are introduced in detail :"In tsavorites with high V content, the                  absorption at 430nm is due to V; The 605nm is consist of 590nm and                    643nm. The absorption peak at 643 nm is attributed to the transition of d            electrons of V3+ from 3T1g ground state to 3T2g and 1T2g states. The                    absorption peak at 590 nm can be attributed to the transition absorption            of 3d3 electrons of Cr3+ between 4A2g → 4T2g energy levels."

     3. Re: R%=Reflectance%, Tsa=tsavorite
